

# A dual-wavelength photothermal aerosol absorption monitor: design, calibration and performance

Luka Drinovec[1,2,3], Uroš Jagodič[1,2], Luka Pirker[2,4], Miha Škarabot[2], Mario Kurtjak[5], Kristijan
Vidović[6], Luca Ferrero[7], Bradley Visser[8], Jannis Röhrbein[8], Ernest Weingartner[8], Daniel M.
Kalbermatter[9], Konstantina Vasilatou[9], Tobias Bühlmann[9], Celine Pascale[9], Thomas Müller[10],
Alfred Wiedensohler[10], Griša Močnik[1,2,3]

[1]Haze Instruments d.o.o., Ljubljana, Slovenia
[2]Department of Condensed Matter Physics, Jozef Stefan Institute, Ljubljana, Slovenia
[3]Center for Atmospheric Research, University of Nova Gorica, Nova Gorica, Slovenia
[4]Faculty for mathematics and physics, University of Ljubljana, Ljubljana, Slovenia
[5]Advanced Materials Department, Jozef Stefan Institute, Ljubljana, Slovenia
[6]Deparment for Analytical Chemistry, National Institute of Chemistry, Ljubljana, Slovenia
[7]GEMMA center, University of Milano-Bicocca, Milano, Italy
[8]Institute for Sensors and Electronics, University of Applied Sciences Northwestern Switzerland,
Windisch, Switzerland
[9]Federal Institute of Metrology METAS, Bern, Switzerland
[10]Leibniz Institute for Tropospheric Research, Leipzig, Germany

*Correspondence to*: Luka Drinovec (luka.drinovec@haze-instruments.com)




**Abstract**

There exists a lack of aerosol absorption measurement techniques with low uncertainties and without artefacts. We have developed a two-wavelength Photothermal Aerosol Absorption Monitor (PTAAM-2λ), which measures the aerosol absorption coefficient at 532 and 1064 nm. Here we describe its design, calibration and mode of operation and evaluate its applicability, limits and uncertainties. The 532 nm channel was calibrated with ~1 μmol/mol $NO_2$, whereas the 1064 nm channel was calibrated using measured size distribution spectra of nigrosin particles and a Mie calculation. Since the aerosolized nigrosin used for calibration was dry, we determined the imaginary part of the refractive index of nigrosin from the absorbance measurements on solid thin film samples. The obtained refractive index differed considerably from the one determined using aqueous nigrosin solution. PTAAM-2λ has no scattering artefact and features very low uncertainties: 4% and 6% for the absorption coefficient at 532 nm and 1064 respectively; and 9% for the absorption Ångström exponent. The artefact-free nature of the measurement method allowed us to investigate the artefacts of filter photometers. Both the Aethalometer AE33 and the CLAP suffer from the cross-sensitivity to scattering – this scattering artefact is most pronounced for particles smaller than 70 nm. We observed a strong dependence of the filter multiple scattering parameter on the particle size in the 100-500 nm range. The results from the winter ambient campaign in Ljubljana showed similar multiple scattering parameter values for ambient aerosols and laboratory experiments. The spectral dependence of this parameter resulted in AE33 reporting the absorption Ångström exponent for different soot samples with values biased 0.17-0.3 higher than the PTAAM-2λ measurement. Photothermal interferometry is a promising method for reference aerosol absorption measurements.

45



## 1 Introduction

Absorption of sunlight by aerosols, especially black carbon, and the related (semi)indirect effects of these aerosols are collectively the second leading cause of atmospheric warming (Bond et al., 2013). The same light-absorbing aerosols have serious detrimental health effects (Janssen et al., 2011; Janssen et al., 2012). They can be directly emitted into the atmosphere (primary aerosols - black carbon and a fraction of the light-absorbing organic aerosols) or produced in the atmosphere from precursor gases (secondary aerosols - some of the light-absorbing organic aerosols) and their optical properties change with ageing (Saleh et al., 2013; Kumar et al., 2018). Their short atmospheric lifetime (compared to $CO_2$) makes them an appealing target for fast abatement and is responsible for their heterogeneity. Climate models do not agree with or capture all the complexity of aerosol absorption observations (Bond et al., 2013; Samset et al., 2018). In order to quantify any reduction in atmospheric heating (Ferrero et al., 2014), aerosol absorption needs to be measured precisely and accurately in many different types of sites, especially regional background sites (Zanatta et al., 2016). Moreover, the measurement of absorption coefficient should be done at different wavelengths for the understanding the direct and semi-indirect aerosol effects on the climate (Samset et al., 2018) and for determining the heating rates, the contribution of sources to these effects and the role of clouds (Ferrero et al., 2021). The increase of absorption at lower wavelengths can be parameterised as an increased absorption Ångström exponent (AAE) compared to AAE=1 exhibited by black carbon. This increase can be due to the presence of absorbing organic aerosol (Kirchstetter et al., 2004) or organic and inorganic coatings on black carbon (Zhang et al., 2018; Virkkula, 2020). The source (or combustion efficiency) specific AAE can be used in source apportionment of black carbon (Sandradewi et al., 2008; Zotter et al., 2017). However, the measurement of the aerosol absorption coefficient is still a challenging task.

The absorption coefficient has been most often determined using the most practical instrumentation – filter absorption photometers: the Aethalometer, the Particle Soot Absorption Photometer (PSAP), the Continuous Light Absorption Photometer (CLAP) and the Multiple Angle Absorption Photometer (MAAP) (Rosen et al., 1978; Drinovec et al., 2015; Bond et al., 1999; Ogren et al., 2017; Petzold et al., 2002), where the measurement of light attenuation in the sample-laden filter relative to the clean filter is converted to mass equivalent black carbon concentration (BC) (Petzold et al., 2013) or the absorption coefficient. This type of measurement requires different assumptions which are hard to verify. The enhancement of absorption in the scattering filter matrix and the mass absorption cross-sections are two external parameters which need to be determined. Additionally, the measurement is non-linear and the reduction of the sensitivity due to filter loading and the cross-sensitivity to scattering are artefacts, the contributions of which are either assumed (Bond et al., 1999; Ogren et al., 2017) or measured to different extents (Petzold et al., 2002; Drinovec et al., 2015). Post-processing algorithms were developed to compensate the loading or concentration effects (Weingartner et al., 2003; Virkkula et al., 2007; Collaud Coen et al., 2010; Hyvärinen et al., 2013). These effects are, to a degree, sample dependent, so the data analysis requires more sophisticated approaches (Drinovec et al., 2015; Drinovec et al., 2017) and an additional measurement of scattering is needed (Arnott et al., 2005; Ogren et al., 2017; Yus et al., 2021) to fully compensate for the sensitivity dependence on the sample single-scattering albedo (SSA) and other artefacts.

In situ methods (i.e. without collecting the sample on a filter) offer advantages over filter photometers, such as separate measurements of aerosol extinction and scattering coefficients. The

integration of these two measurements in a single instrument (Petzold et al., 2013) allows the
measurement of these quantities for the exact same sample and calculation of their difference yields
the absorption coefficient. Thus, this method was used as a reference for filter photometer
characterization (for example, Bond et al., 1999). However, this measurement is still severely
restricted when measuring aerosol exhibiting high SSA, as measurement errors of several percent
can result in absorption coefficient errors of above 100%, which requires adherence to a strict
measurement and data post-processing algorithm to minimize these errors (Modini et al., 2021). On
the other hand, the fractal nature of low-SSA black carbon aerosol and the effect of this morphology
on the scattering truncation correction of the nephelometer present a systematic source of error also
for measurements at low SSA (Modini et al., 2021).

A direct measurement of the aerosol absorption coefficient would therefore avoid the described
issues. This is possible in photoacoustic instruments, in which an air sample is drawn through the
sample chamber, where it is illuminated with a powerful pump laser beam. The light absorbed by the
aerosol is converted to heat, causing an increase of local temperature. Transport of heat causes
changes in density and the acoustic wave is amplified in a chamber that acts as an acoustic
resonator. The pump beam is modulated and phase-sensitive detection is used to amplify the signal
over the measurement noise. Acoustic resonance amplifies the signal by orders of magnitude but
needs to be tracked actively to maintain maximum sensitivity. Moreover, the method experiences
systematic biases when the sample contains semi-volatile organic coatings or water. As these
substances evaporate from the aerosol, the latent heat causes a reduction of the detected acoustic
signal (Arnott et al., 2003; Murphy, 2009; Langridge et al., 2013). A comprehensive review of these
different approaches can be found in Moosmüller et al. (2009).

Photothermal interferometry similarly employs a modulated pump laser to heat the sample with the
absorbed laser light. A second interferometric laser probes the change of the refractive index caused
by light absorption and the subsequent heating and decrease of the density of the sample. The
response of photothermal interferometer (PTI) to the aerosol absorption coefficient is linear. The
first proposed and realized PTI instruments employed a folded Jamin interferometer due to its
inherent mechanical stability and a glancing pumping geometry, where the pump beam and the
probe interferometric beam overlap at an acute angle in the sample (Moosmüller and Arnott, 1996;
Sedlacek, 2006; Sedlacek and Lee, 2007; Lee and Moosmüller, 2020). A novel folded Mach–Zehnder
interferometric design employs a single laser for both heating and interferometric detection (Visser
et al., 2020).

Standardization, validation and calibration of the aerosol absorption coefficient measurement
require well-defined reference samples: particles or absorbing gases (Arnott et al., 2000). The
greatest advantage of in situ methods is the ability of calibration with gases (Arnott et al., 2000; Lack
et al., 2006, 2012; Nakayama et al., 2015; Davies et al., 2018), while filter photometers need to be
compared with each other (Müller et al., 2011; Cuesta-Mosquera et al., 2020) or with other
reference instruments (Bond et al., 1999; Arnott et al., 2003) using model aerosols: fullerene soot,
spark discharge particles, soot generated with controlled gas combustion, and dyes, especially
nigrosin (Bond et al., 1999; Schnaiter et al., 2006; Müller et al., 2011; Cuesta-Mosquera et al., 2020).
The advantage of using gases for calibration is conceptual – gas concentration can be measured with
great precision and accuracy, and its absorption cross-section is usually well known. It is also practical
– gas calibration can be performed in the field, while instruments calibrated with aerosolized





particles need to be sent to a calibration laboratory as this effort requires a sophisticated setup. $NO_2$, which absorbs strongly in the blue-green part of the spectrum, is a commonly used calibration gas.

Calibration with $NO_2$ is very common, especially for climate studies, because it is usually performed at green wavelengths, where the solar spectrum features a maximum. It is also convenient experimentally thanks to the large absorption cross-section of $NO_2$ in this part of the visible spectrum. However, calibration with gases at longer wavelengths (e.g. in the red or infrared parts of the spectrum) is either extremely challenging or impossible. Yet, these regions are important as they allow the determination of the aerosol absorption wavelength dependence over the whole solar

spectrum. This enables a conceptually simple definition of BC, separate from light-absorbing organic aerosols, which feature most of their absorption at lower wavelengths (Kirchstetter et al., 2004; Lack et al., 2008), and the conversion of the optical measurement into BC. To calibrate absorption instruments at longer wavelengths, absorbing particles can be used instead of gases, e.g., Aquadag (Foster et al., 2019), absorbing polystyrene spheres (Lack et al., 2009) and especially nigrosin (Lack et

al, 2006; Bluvshtein et al., 2017; Foster et al., 2019). Nigrosin is water soluble and forms spherical aerosols, which can be well described using Mie theory. It has been shown that polydisperse nigrosin can be used for calibration (Foster et al., 2019).

This work describes a novel photothermal interferometer design, which allows for simultaneous and colocated absorption measurements at two wavelengths. The article has been divided into a number

of sections to provide a thorough treatment of the instrument: firstly the physical construction of the instrument is described, with a special focus on the novel geometry of the pump and probe beams in the sample chamber. The temporal evolution of the signal measured by the photodiodes during the modulation period of the pump beam and the selection of the modulation frequency is then discussed. Subsequently a noise analysis is performed, showing the sources of noise in the

instrument. Following this the operation of the interferometer and the interferometric signal is detailed. A section of the article is then dedicated to the two-wavelength calibration procedure. The linearity of the measurement is demonstrated and the stability and noise of the instrument are determined. Finally, the results obtained with the instrument during a number of laboratory and ambient campaigns are presented.



**2 Materials and methods**

**2.1 Chemicals**

Nigrosin (Acid black 2, Nigrosin water soluble, CAS 8005-03-6) and ammonium sulphate (Mascagnite, ReagentPlus®, CAS 7783-20-2) were obtained from Sigma-Aldrich. Aqueous solutions of nigrosin (labelled, N1-N4) and ammonium sulphate (AS1-AS3) were prepared by dissolving chemicals in ultrapure water (Milli-Q) (Table 1).

**Table 1. Designations of the aqueous solutions of nigrosin (N1-N4) and ammonium sulfate which were nebulized.**

| Sample | Concentration (g/l) |
|--------|---------------------|
| N1 | 0.01 |
| N2 | 0.1 |
| N3 | 1 |
| N4 | 10 |
| AS1 | 0.01 |
| AS2 | 0.1 |
| AS3 | 1 |

Solid nigrosin samples were prepared on microscope slides by drying the aqueous solutions of nigrosin (see sample photographs in Supplement S5.3). Glass slides were first cleaned with isopropyl alcohol, then a hydrophilic layer was produced by dipping the slide into a 4% detergent solution (Hellmanex III) for 30 seconds and flushing the slide with Milli-Q water. Finally 0.2 ml of nigrosin solution was spread on a surface of approx. 6 cm$^2$ and left to dry for 12 hours.

Two bottles of 1 μmol/mol $NO_2$ in synthetic air were acquired (Traceline, Messer Schweiz AG) for the calibration. Calibration of the photothermal aerosol absorption monitor (PTAAM-2λ) at 532 nm was performed by filling a 10 l Tedlar® SCV Gas Sampling Bag (Sigma-Aldrich) with the calibration gas and connecting it to the inlet of the instrument.

**2.2 Generation of $NO_2$ reference gas mixture**

Poor stability of reference gas mixtures in the nmol/mol range in pressurized cylinders is a well-known issue for $NO_2$ due to its reactivity (USEPA, 2017; Flores et al., 2021). To generate dynamically SI-traceable $NO_2$ reference gas mixtures at low amount fractions in the field, METAS developed the "traceable mobile permeation generator" (TMPG), which is based on the permeation method (Haerri et al., 2017). The main components of the TMPG are a temperature stabilized permeation chamber, with a calibrated temperature sensor in the chamber, and a calibrated mass flow meter. Calibrated permeation units are placed in the permeation chamber. At a constant temperature, the permeation rate, which here is the transfer of $NO_2$ from the reservoir of the permeation device through its membrane, becomes stable after a stabilization time of at least 24 h. After stabilization, the $NO_2$ reference gas mixture is continuously generated at the desired amount fraction by setting the flow of the dilution gas.

First, the permeation rate of the permeation unit (produced by Fine Metrology S.r.l.s., Italy) containing high purity $NO_2$ was calibrated in a magnetic suspension balance (MSB; Swiss primary



reference system) at 40 and 45 °C at an absolute pressure of 1080 mbar in synthetic air 5.6 for several days. The permeation unit was then placed in the permeation chamber of the TMPG that mimics the conditions in the MSB. TMPG was sent to Ljubljana. After 4 days of stabilization the generated gas was used for calibration of PTAAM-2λ and CAPS NO$_2$ monitors. After shipment back to METAS the permeator was again analyzed in MSB.

**2.3 Laboratory aerosol generation**

Nigrosin and ammonium sulfate aerosols were generated using the ATM 226 nebulizer (Topas GmbH, Germany) set to 4 lpm sample flow. The nebulizer was connected to diffusion driers to reduce relative humidity below 30%.

**2.4 Soot samples**

Diesel exhaust was collected from the tailpipe of a EURO3 Volkswagen Passat 1.9 TDI at 2500 rpm.
An average black carbon emission factor of 1.31 g/kg was previously determined for the selected vehicle (Ježek et al., 2015).

Soot generated by a miniCAST 5201 Type BC (Jing Ltd., Switzerland) was also used as a test aerosol. The physical and optical properties of the soot particles have been reported elsewhere (Ess et al., 2021).

Propane soot samples were obtained by collecting the exhaust of a portable propane burner/torch. After ignition the air inlets were closed to generate fuel-rich combustion. Because of slow sample dilution large soot agglomerates were formed.

**2.5 Absorbance measurements**

Absorbance of nigrosin was measured using a Shimadzu UV-3600 UV-VIS-NIR Spectrophotometer
(Shimadzu, Japan). Aqueous solutions of nigrosin were measured in the 10-mm-path quartz cuvette. Solid nigrosin samples were placed at the entrance of the ISR-3100 integrating sphere, which collected all the transmitted illumination.

**2.6 Reflectance measurements**

Reflectance of nigrosin using a perpendicular beam at 520 and 1064 nm was measured to correct for
the reflection losses during the absorbance measurements. Reflectance was calculated by dividing the intensity of the reflected beam with that of the incident beam.

To determine the real part of the refractive index, a measurement of the Brewster θ$_B$ angle was performed at 633 and 1064 nm. Measurement was performed on a thick film of nigrosin on a glass slide.

**2.7 Online instrumentation**

The size distribution of the aerosol particles was measured using a scanning mobility particle sizer (TSI model 3936L75). The instrument was used with an impactor with a 0.0508 cm nozzle. Measured spectra were corrected using the diffusion and the multiple charge correction algorithams. Scattering was measured with the Aurora 4000 – Polar Nephelometer (Ecotech, Australia) set to measure the
total and back scattering coefficient.



Two filter photometers, an Aethalometer model AE33 (Magee Scientific, USA) and a CLAP (Haze Instruments, Slovenia) were used to obtain attenuation coefficients. The AE33 features a built-in filter loading compensation algorithm (Drinovec et al., 2015) and was using the M8060 filter. The CLAP was run using the Azumi 371M filter and the data have been compensated manually using the algorithm by Ogren et al. (2017).


Two $NO_2$ monitors based on cavity ring-down spectroscopy were used to determine $NO_2$ amount fraction. In 2020, a $NO_2$ analyzer – model T500U (Teledyne, USA) was used. In 2021, the CAPS $NO_2$ monitor (Aerodyne research, USA) was used.

### 2.8 Offline instrumentation

To determine the morphology of the diesel exhaust, propane soot, and nigrosin samples, a FEI HeliosNanolab 650 (Thermo Fischer Scientific, Waltham, MA, USA) scanning electron microscope (SEM) operating at 1 kV accelerating voltage was used. The samples were mounted on SEM holders with carbon tape and coated with a few nm of carbon to prevent charging effects.

To determine the laser stability during warmup and operation, the spectra of the 532 nm DPSS laser
were measured with a spectrometer (Shamrock SR-500i, Andor) equipped with a cooled EMCCD camera (Newton DU970N, Andor).

An atomic force microscope (AFM) Nanoscope IIIa - MultiMode AFM (Digital Instruments, Santa Barbara, CA) equipped with J scanner (100 µm horizontal range) operating in contact mode was used to determine the thickness of the solid nigrosin samples. Measurements were performed by carving
a groove in the thin layer of the nigrosin sample all the way down to the glass substrate and measuring the depth of the groove.

### 2.9 Mie calculation

Mie calculations were performed using Matlab routines for homogeneous spheres (Mätzler, 2002).

### 2.10 Measurement campaigns

The presented data was collected during several measurement campaigns:
- Ljubljana ambient winter campaign 2020: duration: 26 Feb 2020 – 24 Mar 2020; instrumentation: the PTAAM-2λ and the AE33.
- Ljubljana laboratory campaign 2020: first PTAAM-2λ characterization campaign; instrumentation: Aurora 4000, AE33, CLAP, Teledyne $NO_2$ monitor;
- AeroTox laboratory campaign 2020: study on the influence of coating on absorption (Kalbermatter et al., 2021); PTAAM-2λ and SMPS measurements of the uncoated CAST soot and nigrosin are used here
- Ljubljana laboratory campaign 2021: additional PTAAM-2λ characterization measurements; instrumentation: Aurora 4000, AE33, CLAP, Aerodyne research $NO_2$ monitor; SMPS.


### 3 The instrument description

### 3.1 The instrument setup

The dual-wavelength photothermal aerosol absorption monitor (PTAAM-2λ) uses the photothermal effect to measure aerosol absorption. A folded Mach-Zehnder interferometer is used to measure the
difference of the optical path between the two interferometer beams (Figure 1). For the probe beam a HeNe laser is coupled into a polarisation-maintaining single mode fiber. The beam exits the fiber via a fiber collimator providing a 0.25 mm diameter free-space beam, which is split into two interferometer beams within a beam-splitter block. The returning beams are combined and the intensity is measured using silicon photodiodes PD1 and PD2 connected to the lock-in amplifier
(Zurich instruments, model MFLI).

The sample cell contains pressure, temperature and relative humidity sensors. The interferometer is sealed in the enclosure to protect it from acoustic noise and pressure fluctuations. The optical path length of one of the interferometer beams is controlled by a pressure cell connected to a computer-controlled syringe pump.

A 2 W frequency-doubled Nd:YAG laser (532 nm) and a 3 W Nd:YAG (1064 nm) laser are coupled to the multimode optical fibers. Parts of these fibers are shaken by an electromagnetic actuator to homogenise the profiles of the beams exiting the fibers. The pump beams, which are modulated at different frequencies, are collimated and introduced by dichroic mirrors into the interferometer. The powers of the pump beams are monitored with photodiodes P1 and P2.

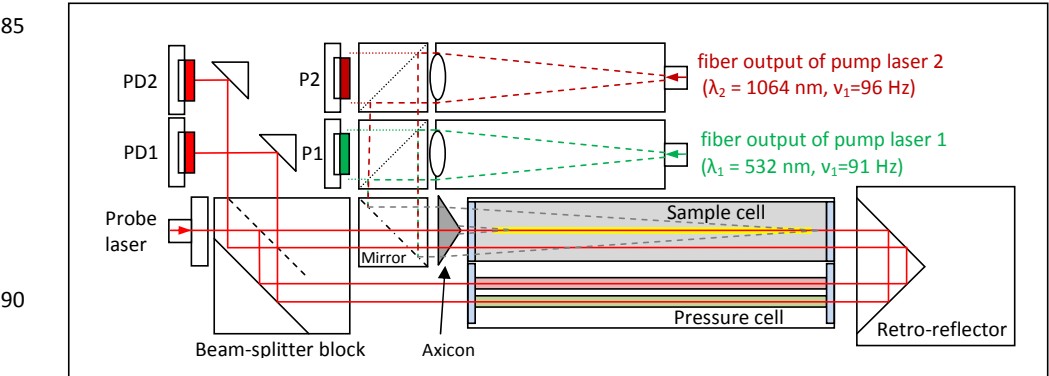


**Figure 1. Schematic representation of the PTAAM-2λ setup.**





A new photothermal interferometer configuration is utilised which uses an axicon to focus the pump
beams into the sample chamber where they overlap with the probe beam (Drinovec and Močnik,
2020). The pump beams exiting the fibers are collimated and subsequently combined using two
dichroic mirrors and then directed into the measuring chamber using a custom drilled mirror – two
drilled holes allow the interferometer beams to pass through the mirror without interacting with the
pump beam. A custom axicon, drilled to allow passage of the probe beams, is used to provide an
elongated focus of the pump beams along the axis of the probe beam inside the sample cell. The
diameters of the probe and pump beams in the sample cell are shown in Figure 2. The pump beams
are aligned with the probe beam using a CCD camera. During one and a half years of instrument
testing (including road shipment in excess of 3000 km to the two measurement campaigns) there
was no need to realign the optics.

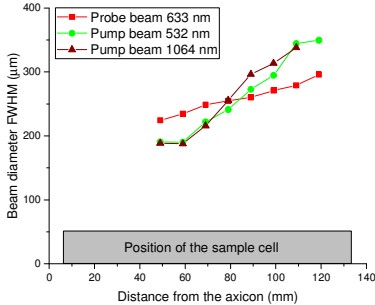


**Figure 2. Beam diameters of probe and pump beams inside the sample cell. Beam sizes closer to the axicon could not be measured due to mechanical restrictions.**

This setup allows for simultaneous measurement of absorption at two wavelengths in the same
sample volume by modulating the probe beams at different frequencies (91 Hz and 96 Hz, for
example). The retention time of the aerosol particles in the pump beam (approx. 5 seconds for a 100
nm particle) is much longer compared to the duration of the modulation interval (11 ms).

For "in situ" aerosol absorption measurement the pump beam intensity should be kept low enough
so that the heating does not modify the physical properties of the particles. The use of an axicon for
focusing the pump beams results in a pump illumination intensity of approx. 2 W/mm$^2$, which is
much lower compared to the glancing setup (Sedlacek, 2006) and the modulated single-beam PTI
(Visser et al., 2020).


### 3.2 Photothermal effect

When the sample air is illuminated with the pump beam, some of the light is absorbed by the gases
and particles in the sample. Heat is then transferred to the surrounding air causing an increase of
temperature. This, in turn, causes a small reduction in the air refractive index. This change of the
refractive index is measured as the reduction of the optical path in the folded Mach-Zehnder
interferometer.

The pump beam is modulated ON/OFF with a 50% duty cycle. When the pump beam is switched ON,
there is first a linear increase of the photodiode voltage which later starts to saturate due to
increasing heat transfer to the surrounding air (Figure 3). The exact shape of the detected voltage
depends on the intensity profile of the pump beam during the heating period and the geometry of
the pump and probe beams. For the pump frequencies between 90 and 100 Hz, we observed an
exponential rise and decay of the detector voltage during the ON and OFF part of the modulation
period. Here, we show the results obtained for nigrosin using pump lasers operating at 532 nm and
1064 nm. The amplitude of the 532 nm channel voltage is approximately 7-fold higher compared to
the 1064 nm channel, but the shape does not change with amplitude. Therefore, nonlinearities in the
instrument response can be avoided and a lock-in amplifier can be used to measure the amplitude of
the photodiode voltage.

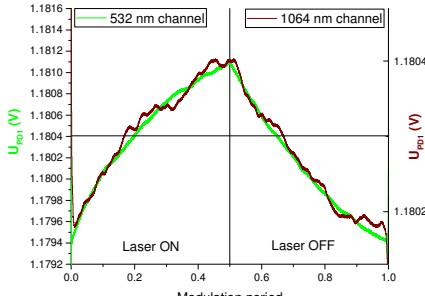

340

**Figure 3. Average photodiode voltage during the heating and cooling parts of the modulation
period for the 532 nm and the 1064 nm channels obtained for nigrosin particles. The duration of
the modulation period is 11.1 ms and 10.4 ms, respectively. The curves are obtained by averaging
the photodiode PD1 voltage over more than 10000 modulation periods.**

### 3.3 Investigation of photothermal signal and noise

When the optical path of the interferometer changes, we observe a voltage increase on one of the photodiodes and a decrease on the other. To increase the signal-to-noise ratio, the difference between the photodiode voltages ($U_{PD1}$-$U_{PD2}$) is used to calculate the photothermal signal. This difference is fed to the lock-in amplifier using the pump beam modulation frequency as the reference frequency. The lock-in amplifier multiplies the signal with a sine/cosine function at the modulation frequency and filters out other frequencies; this results in two output signals, $U_x$ and $U_y$, which represent the dynamic components phase-shifted by $\pi/2$. The highest lock-in signal for the selected modulation frequency is obtained at a certain phase $\beta$, being approx. 2 rad for a modulation frequency of 91 Hz. The projection of the lock-in signal is calculated as a scalar product:

$$S = U_x cos(\beta) + U_y sin(\beta). \tag{1}$$

The instrument response S strongly depends on the duration of the modulation period (Figure 4). The signal increases with the duration of the modulation period, but shows a saturation effect because of the increased heat transfer. In general, the measurement noise (defined as standard deviation of the 1 s data) also increases with the length of the modulation interval. The optimum modulation frequency is selected by the best signal-to-noise ratio.

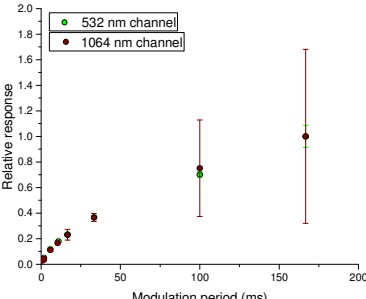

**Figure 4. Instrument response S (lock-in amplifier signal at the optimum phase) at different durations of the modulation period. The responses are normalised to the response at 167 ms. The error bars designate a standard deviation of 1 s data.**

Increased noise at lower frequencies can also be observed on fast Fourier transformation of the photodiode signal (Figure S3). Noise is highest below 10 Hz, there is a plateau between 10 and 100 Hz, and increased noise between 200-300 Hz. There are also certain noise peaks, which should be avoided to reduce the measurement noise. Frequencies of 91 and 96 Hz were selected to attain the highest signal-to-noise ratio.

When there is no absorbing sample in the measurement cell, it is possible to measure an offset with the magnitude of 8 µV for the 532 nm channel and 39 µV for the 1064 nm channel. This offset is generated in phase with the measurement frequency. The main sources of the offset are electronic interaction (signal at the pump frequencies is leaked into the photodiode signal) and interaction of the pump beam with the probe beam on the shared optical elements. Variation of the offset is an additional source of measurement noise.





There are several sources of noise which are independent of the pump frequency, including the noise of probe lasers, photodiodes, sample air flow turbulences, pump laser power oscillations, mechanical resonances of the optical elements and random electronic noise. The sources of noise were investigated by comparing different experimental setups (Figure 5):

- "PD": pump and probe lasers switched OFF; noise is generated by photodiodes and electronics.

- "HeNe": HeNe probe laser is switched ON with one of the interferometer arms blocked; measured noise comes from the variation of the HeNe laser power.

- "Interferometer": HeNe probe laser is switched ON; this is a measure of the interferometer noise.

- "Interferometer + flow": HeNe probe laser is switched ON, sample flow (0.4 lpm) is switched ON;
there is some increase of noise due to the airflow in the sample cell.

- "Interferometer + pump beam": Probe and pump lasers and sample flow are switched ON; the noise coming from offset variation is added.

The results show that the majority of the noise comes from the interferometer. It may be related to the vibration of the optical elements of the interferometer. For the 1064 nm channel about one half
of the noise is caused by the offset variation.

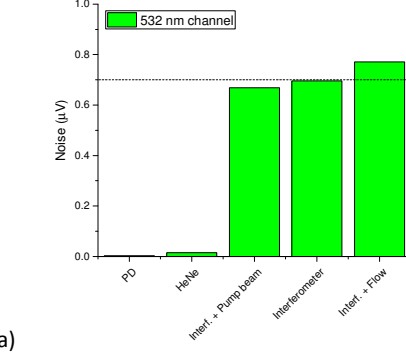

a)

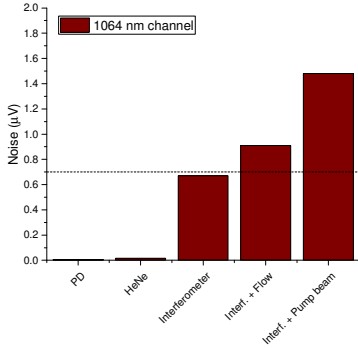

b)

**Figure 5. Contribution of different sources to the measurement noise (standard deviation of signal) for the 532 nm channel (a) and the 1064 nm channel (b) run at 91 and 96 Hz, respectively. Settings: "PD" – photodiode noise, "HeNe" – noise of the HeNe laser (blocked one of the interferometer**
**arms), "Interferometer" – noise of the interferometer, "Interf. + Flow" – interferometer noise for sample flow switched on, "Interf. + Pump beam" – interferometer with a pump beam switched on.**



### 3.4 Calculation of the PTAAM-2λ signal

The response of photothermal interferometer is highest when the interferometer is operating in the so-called quadrature point – where the difference of the optical paths between the two arms of the interferometer is exactly λ/4. The interferometer response in the quadrature point is linear, as photodiode voltages in both interferometric arms intersect at the steepest (and linear) part of the sine curve (Figure 6). The quadrature point can be determined by performing an interferometric scan

(measurement of the photodiode signal while changing the optical path difference). The pressure cell is used to regulate the optical path of one interferometer arm to maintain the interferometer in the quadrature point.

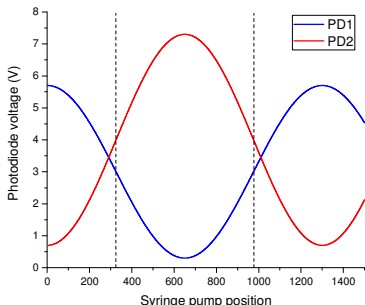

**Figure 6. The interferometer scan of photodiode PD1 and PD2 voltages is performed by changing the optical path of one of the interferometer arms using the pressure cell. A 1λ difference in optical**

**path corresponds to 1300 steps of the syringe pump. Quadrature points are marked by dashed lines.**

For an interferometer in the quadrature point, the lock-in signal is proportional to the amplitude of the interferometer signal $U_{I0}$. It is determined from the maximum and minimum voltages for both photodiodes determined during the interferometer scan:

$$U_{I0} = \frac{U_{PD1max} - U_{PD1min} + U_{PD2max} - U_{PD2min}}{4}. \tag{2}$$

Because the HeNe laser intensity slowly changes with time, a special procedure to determine the instantaneous interferometer amplitude $U_I$ was developed, which holds when the interferometer is in the quadrature point:

$$U_I = \left(\frac{U_{PD1} + U_{PD2}}{U_{PD10} + U_{PD20}}\right) U_{I0}. \tag{3}$$

$U_{PD1}$ and $U_{PD2}$ are the instantaneous photodiode signals in the quadrature point, and $U_{PD10}$ and $U_{PD20}$ the photodiode signals in the quadrature point when the interferometer scan was performed. The projections of the lock-in signals for the two channels are:

$$S_{532nm} = U_{x,532nm} cos(\beta_{532nm}) + U_{y,532nm} sin(\beta_{532nm}), \tag{4}$$

$$S_{1064nm} = U_{x,1064nm} cos(\beta_{1064nm}) + U_{y,1064nm} sin(\beta_{1064nm}). \tag{5}$$

The absorption coefficient is calculated by dividing S with the instantaneous interferometer amplitude $U_I$, reported at the selected pressure $p_0$ and temperature $T_0$, where $p$ and $T$ represent instantaneous conditions. The response is adjusted by the calibration constant A and the ratio between the nominal $P_{\lambda,0}$ and instantaneous pump laser power $P_\lambda$, separately for each channel:

$$b_{abs,532nm,total} = A_{532nm}\left(\frac{S_{532\,nm}}{U_I}\right)\left(\frac{p_0}{p}\right)\left(\frac{T}{T_0}\right)\left(\frac{P_{532nm,0}}{P_{532nm}}\right),$$ (6)


$$b_{abs,1064nm,total} = A_{1064nm}\left(\frac{S_{1064\,nm}}{U_I}\right)\left(\frac{p_0}{p}\right)\left(\frac{T}{T_0}\right)\left(\frac{P_{1064nm,0}}{P_{1064nm}}\right).$$ (7)

The baseline offset is determined for the filtered air sample and subtracted from the total signal to provide the absorption coefficients $b_{abs,532nm}$ and $b_{abs,1064nm}$:

$$b_{abs,532nm} = b_{abs,532nm,total} - b_{abs,532nm,offset},$$ (8)

$$b_{abs,1064nm} = b_{abs,1064nm,total} - b_{abs,1064nm,offset}.$$ (9)





**4 Calibration of the PTAAM-2λ**

The instrumental response depends on the overlap between probe and pump beams. After beam
alignment, which proved very stable over a very long period, both channels need to be calibrated.

**4.1 Calibration of the 532 nm channel**

Similarly to the calibration of photoacoustic instruments (Arnott et al., 2000; Lack et al., 2006;
Bluvshtein et al., 2017), we decided to use $NO_2$ for calibration of the 532 nm channel. An absorption
cross-section of $1.47*10^{-19}$ $cm^2$ was determined from the measured laser spectrum and the high-
resolution absorption spectrum of $NO_2$ by Vandaele et al (2001) (Supplement S4). An absorption
coefficient of 1 μmol/mol $NO_2$ at 100 kPa and 25 °C of 357.3 $Mm^{-1}$ was obtained.

Calibration of the 532 nm channel was performed by filling the Tedlar bag (Sigma-Aldrich) with $NO_2$
from the bottle of nominally 1 μmol/mol $NO_2$ in synthetic air (Messer Schweiz AG). This procedure
features much lower noise compared to connecting the instrument inlet directly to the bottle. The
Tedlar bag was filled with $NO_2$ and emptied several times to passivate the bag's inner surface. After
the passivation procedure the $NO_2$ amount fraction in the bag agreed within 1 % with the source
concentration. During the offset measurement, a second Tedlar bag, filled with synthetic air, was
sampled.

The measured $NO_2$ amount fraction of different bottles with nominal amount fraction of 1 μmol/mol
differed up to 21%. To reduce this uncertainty we performed the calibration with Traceable Mobile
Permeation Generator" (TMPG) developed by METAS (Haerri et al., 2017). The generated amount
fraction of $NO_2$ during the campaign in Slovenia was 292 nmol/mol with the expanded measurement
uncertainty of +12 and -3 nmol/mol (k = 2). This value was used to determine the accurate amount
fractions of $NO_2$ in the bottles used for calibration. The calibration parameter $A_{532nm}$ was calculated
taking into account the measured losses of $NO_2$ and particles in the instrument sample lines.

**4.2 Calibration of the 1064 nm channel**

Calibration of the 1064 nm channel is more complicated as there are no appropriate absorbing gases
available at that wavelength. Particles with known optical properties can be used here to transfer the
calibration from the $NO_2$ calibrated 532 nm channel to 1064 nm as proposed by Arnott et al. (2000)
and Foster et al. (2019). We decided to use water soluble nigrosin (Acid black 2, CAS 8005-03-6)
because it forms spherical particles when nebulized and dried (Supplement S6). To calibrate the 1064
nm channel, the absorption ratio $b_{abs,1064nm}/b_{abs,532nm}$ needs to be calculated. The absoprtion ratio is
determined using Mie calculations based on:

- sphericity of nigrosin particles confirmed by scanning electron microscopy (Supplement S6),

- nigrosin particle size distribution measured with SMPS,

- the complex refractive index of solid nigrosin at 532 nm and 1064 nm.

With respect to the nigrosin refractive index investigation, the imaginary part can be determined

from its absorbance. Absorbance measurements of both aqueous nigrosin solution and nigrosin film were conducted (Supplement S5). The absorbance of the aqueous solution of nigrosin was measured in a 1-cm-path cuvette. Solid nigrosin samples were produced by drying nigrosin solution on the microscope slides. The absorbance of the solid nigrosin film on these slides was measured with an integrating sphere spectrometer. The results show large (up to ± 30%) differences in the imaginary

part of the refractive index between the aqueous and solid nigrosin (Figure 7). This means that calculating absorption using aqueous solution data can lead to substantial systematic errors. The imaginary part of the solid nigrosin refractive index at 532 nm was determined to be 0.223 which is somewhat lower compared to 0.26 obtained by Bluvshtein et al., (2017). This difference can be attributed to the variability in the nigrosin quality provided by the manufacturer or the measurement

method.

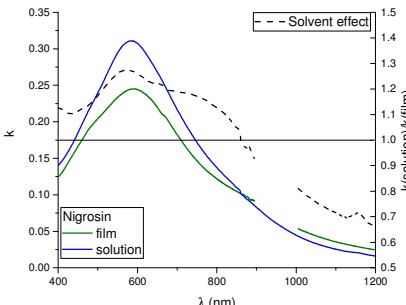

**Figure 7. The imaginary part of the refractive index k determined for solid nigrosin film and its aqueous solution. The solvent effect, the ratio of the determined imaginary parts of the refractive index, is marked by the dashed line.**

Since the real part of the refractive index in infrared has not yet been reported, a measurement of the Brewster angle at 1064 nm was performed (Supplement S5.1) and determined the real part of the refractive index to be 1.848 ± 0.005. We also determined the refractive index at 633 nm to be 1.81 ± 0.01, which agreed well with the value of 1.78 obtained by Bluvshtein et al. (2017). This allowed us to use the real part of the refractive index at 532 nm from Bluvshtein et al. (2017). The

following values of the refractive index have been used for Mie calculations:

$n(532\ nm) = 1.64 + 0.223i$

$n(1064\ nm) = 1.85 + 0.0419i.$



Nigrosin particles were generated using the Topas ATM 226 nebulizer with a 0.1 g/l nigrosin solution
(sample N2) for which we obtained a number size distribution spectrum with a peak at 47 nm. Mie
calculation was performed for each size channel separately and summed together to obtain the
absorption coefficients at 532 and 1064 nm. The calculated absorption ratio $b_{abs,1064\,nm}/b_{abs,532\,nm}$ for
different particle sizes is quite flat between 15 and 300 nm (Figure 8.a). Calculation of the absorption
ratio for different polydisperse distributions shows it to be stable for a wide range of sizes: for
aerosolized nigrosin with a number mode size between 30 nm and 60 nm there is only 2% variation
of the absorption ratio (Figure 8.b). This shows that the calibration with nigrosin is quite a robust
procedure not depending greatly on the nigrosin size distribution.

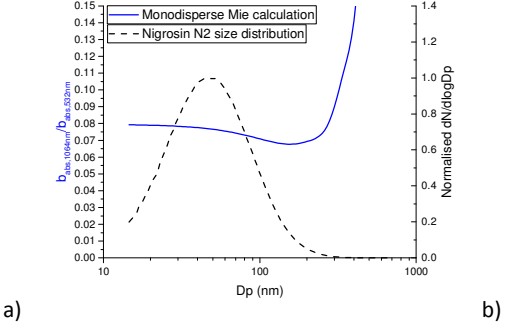
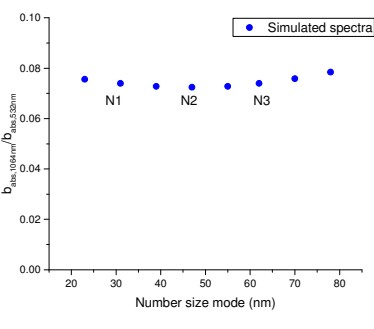

a)

b)

**Figure 8. a) The absorption ratio** $b_{abs,1064\,nm}/b_{abs,532}$ **calculated for monodisperse nigrosin (blue) and
an example of the size distribution for the aerosolized nigrosin solution N2. b) Modelled
absorption ratio calculated for various log-normal size distributions. The number size mode of
aerosolized nigrosin samples N1, N2 and N3 is marked.**

During different measurement campaigns the absorption ratio was calculated for the measured
nigrosin size distributions. Due to the increased measurement uncertainties for particles bigger than
400 nm these data was not used for Mie calculation. Absorption ratio obtained with sample N2
ranged between 0.0733 and 0.0764 (Supplement S3.5). The calibration parameter $A_{1064nm}$ was
adjusted so that the measured absorption ratio agrees with the calculated value.


### 4.3 Validation of Mie calculation at 532 nm

The 532 nm channel of the PTAAM-2λ is calibrated using $NO_2$, which allows the validation of the Mie calculation of nigrosin particle optical properties at that wavelength. The comparison was performed for several nigrosin measurements during the AeroTox 2020 and Ljubljana 2021 campaigns (Table 2). For the AeroTox campaign, the average Mie calculated absorption was 6% while scattering was 17% higher compared to the measurement. For the Ljubljana 2021 campaign, the average Mie calculated

absorption was 24% higher compared to the measurement and the scattering was 26% higher. The biases of Mie calculated absorption and scattering are similar, indicating a similar source for both parameters – that the uncertainty of the SMPS measurements might play a role. Based on our experience, we believe that the absorption coefficient at 532 nm measured by PTAAM-2λ agrees with the Mie calculation based on particle size distributions measured by a well maintained SMPS

instrument.

**Table 2. Mie calculated and measured absorption and scattering coefficients at 532 nm for aerosolized nigrosin N2 for several independent experiments (designated A, B, C) during the AeroTox 2020 and Ljubljana 2021 campaigns. Size distribution below 400 nm was used for Mie calculation. Experiment mean values and standard errors are presented.**

|  | babs_Mie (Mm-1) | babs_meas. (Mm-1) | babs Mie/meas. | bscat_Mie (Mm-1) | bscat_meas. (Mm-1) | bscat Mie/meas. |
|---|---|---|---|---|---|---|
| AeroTox 2020_A | 692 +/- 4 | 628.6 +/- 0.3 | 1.10 | 311 +/- 4 | 269 +/- 1 | 1.16 |
| AeroTox 2020_B | 494 +/- 2 | 463.6 +/- 0.3 | 1.07 | 260 +/- 2 | 214 +/- 1 | 1.21 |
| AeroTox 2020_C | 705 +/- 4 | 696.2 +/- 0.5 | 1.01 | 378 +/- 5 | 329 +/- 2 | 1.15 |
| Ljubljana 2021_A | 597 +/- 6 | 431 +/- 1.3 | 1.38 | 274 +/- 1 | 198.8 +/- 0.3 | 1.38 |
| Ljubljana 2021_B | 625 +/- 4 | 508 +/- 0.6 | 1.23 | 295 +/- 1 | 242.7 +/- 0.3 | 1.22 |
| Ljubljana 2021_C | 537 +/- 7 | 505 +/- 0.6 | 1.06 | 253 +/- 1 | 214.0 +/- 0.3 | 1.18 |




## 5 Results

### 5.1 PTAAM-2λ stability and noise

Before conducting the measurement campaigns, the PTAAM-2λ was calibrated using ~1 µmol/mol
$NO_2$ gas and aerosolized nigrosin particles. The same procedure was used to validate the instrument
sensitivity during the Ljubljana 2020 campaign (Figure 9) assuming a stable $NO_2$ amount fraction and
nigrosin absorption ratio. The 532 nm channel was very stable with 1% variation during one week of
the campaign. For the 1064 nm channel the variation was higher at 3% and showed a slow decline. A
very stable instrument sensitivity was also obtained during the AeroTox campaign (Kalbermatter et
al., 2021).

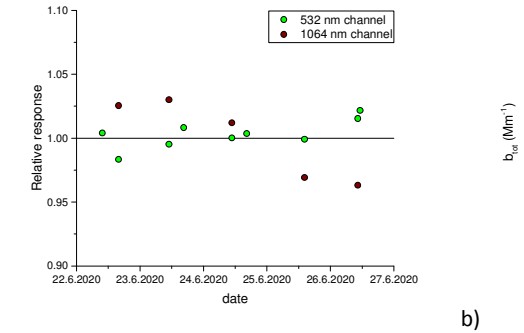  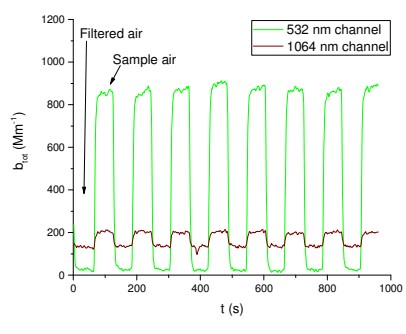

a)                                                                              b)

**Figure 9. Validation of the PTAAM-2λ sensitivity during the Ljubljana 2020 campaign. Validation
was performed by repeating the calibration experiments. Figure b shows the signal during nigrosin
calibration. To obtain the absorption coefficient of nigrosin, the offset obtained for filtered air was
subtracted from the sample air signal.**



Absorption measurements with the PTAAM-2λ are performed by first measuring the filtered air to determine the baseline offset, with the procedure being performed automatically using the internal HEPA filter (Figure 9.b). This offset is later subtracted from the measurement of the sample air.

During the Ljubljana 2020 campaign we measured offsets of 29 Mm$^{-1}$ and 145 Mm$^{-1}$, and standard deviations of 6 Mm$^{-1}$ and 7 Mm$^{-1}$ for the 532 nm and the 1064 nm channels, respectively (Figure 10.a). When applying an additional external HEPA filter, the calculated absorption signal was close to zero with similar standard deviation as observed for the offset.The noise of the instrument was analysed in detail by calculating the Allan variance. The Allan plot of the instrument noise shows a

peak which is characteristic for the smoothed data (Figure 10.b) and is related to the filtering functions used in the lock-in amplifier. For the 532 nm channel we observe a linear decrease of the Allan variance with τ. For the 1064 nm channel the slope is lower due to noise caused by the slow variation of the offset.

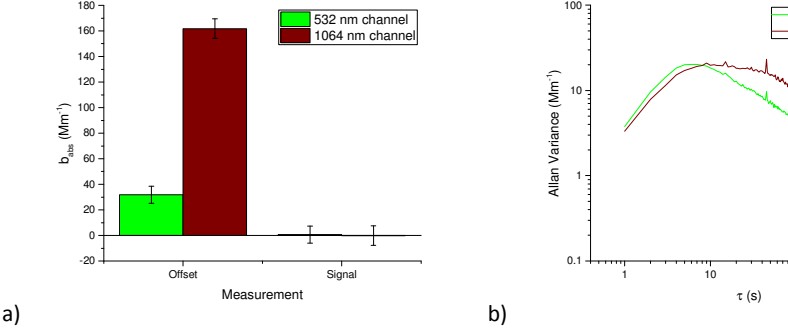

**Figure 10. Average value and standard deviation of the offset and signal for filtered air (a); Allan variance (b).**





**5.2 PTAAM-2λ response linearity for gas and particles**

The linearity of the instrument response was tested with absorbing gases and particles. For $NO_2$
samples with amount fraction below 1 µmol/mol we observed a linear response for the 532 nm
channel with a $R^2$ value of 0.9996 (Figure 11.a). The measurement at 1064 nm confirms that $NO_2$
does not absorb at this wavelength. The linearity was also tested using nigrosin particles. Since the
size distribution of the particles was stable, we were able to use the scattering coefficient as a
reference parameter proportional to the particle concentration. The absorption of nigrosin particles
was linear with the scattering signal in the absorption range up to 1200 $Mm^{-1}$ for the 532 nm channel
and up to 100 for the 1064 nm channel (Figure 11.b), with the corresponding $R^2$ values of 0.997 and
0.968, respectively.

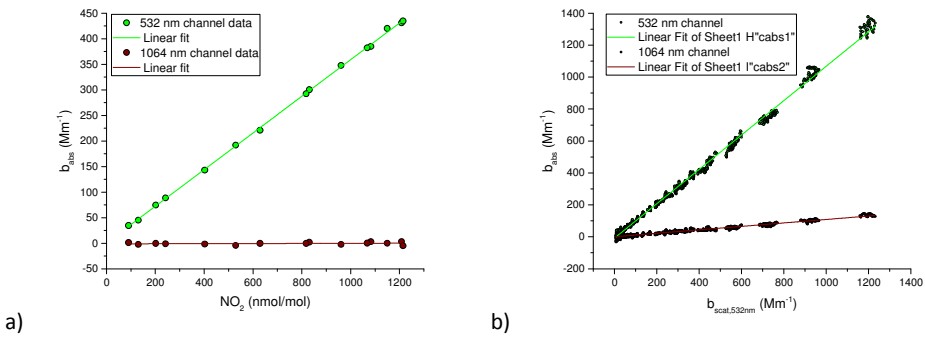

**Figure 11. Instrument response to different amount fractions of $NO_2$ (a) and nigrosin particles (b).**

Interestingly, the noise behaviour for gas measurements differs substantially from that of particle
measurements (Figure 12). There is no significant increase in noise (standard deviation at 1 s time
resolution) with the $NO_2$ amount fraction. For nigrosin particles the noise increases linearly with the
absorption signal and converges toward 1% of the signal. For $NO_2$, light is absorbed directly by the
gas, in contrast to nigrosin, for which heat is transferred from the particles to the gas. This may result
in transient temperature inhomogeneities in the sample volume.

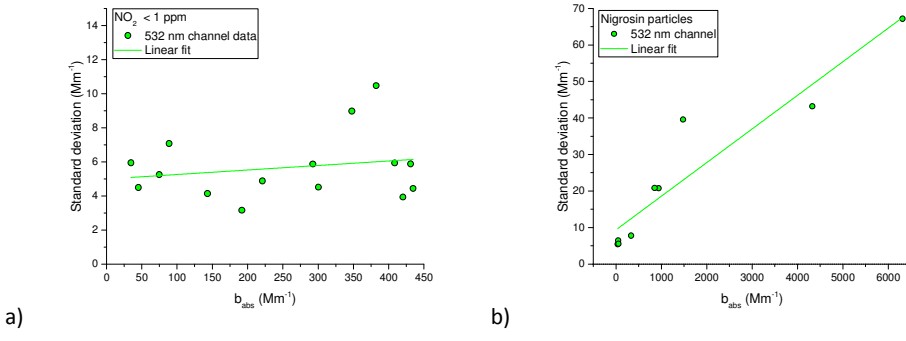

**Figure 12. Standard deviation of the 532 nm channel data during measurement of $NO_2$ (a) and
nigrosin (b).**





### 5.3 PTAAM-2λ response to scattering and absorption

Because scattering does not produce thermal effects, we do not expect photothermal instruments to be sensitive to the scattering of the sample. We generated purely scattering aerosol by nebulizing different aqueous solutions of ammonium sulfate (Figure 13). For the 532 nm channel, we observed a constant signal of about 1.5 Mm$^{-1}$ above the background. This might be caused by the presence of absorbing gases; this absorption is subtracted by the measurement of the filtered air during the

offset determination, but the filtering might also remove a small part of the absorbing gases due to their adsorption on the filter fibres. For the 1064 nm channel, we observe a small negative signal which represents 0.2% of the reported scattering coefficient.

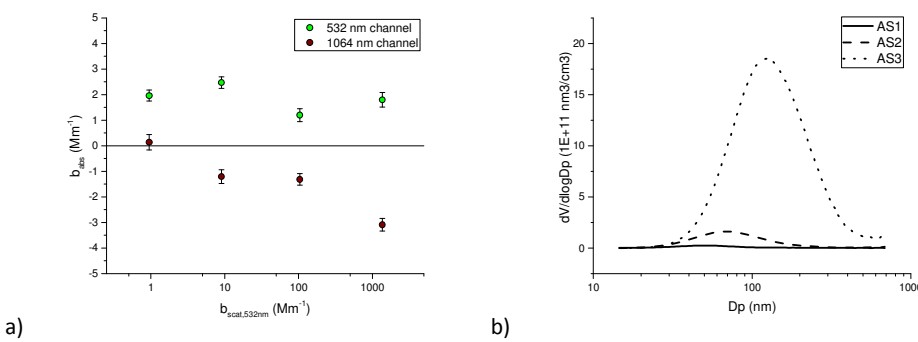

a)    b)

**Figure 13. Scattering artefact for different samples of ammonium sulfate (a). Volume size**
**distributions of ammonium sulfate samples (b).**

**5.4 Laboratory campaigns**

As shown, the PTAAM-2λ measures the aerosol absorption coefficient without size and scattering artefacts. This makes the instrument suitable to study optical properties of aerosols and the

response of other absorption instruments, especially filter photometers. Thus, we have investigated filter artefacts in the Aethalometer AE33 and the CLAP. The response of the AE33 and the CLAP to purely scattering aerosol was tested using aerosolized ammonium sulfate of different sizes (Figure 14.a). The scattering artefact is most pronounced for small particles (volume size mode of 50 nm) where the attenuation coefficient in green approximately equals the scattering coefficient. For larger

particles it is reduced and reaches 6% at particle volume size mode of 122 nm. These values are higher compared to the ones in Drinovec et al. (2015) where $b_{atn}/b_{scat}$ values between 1.2% - 3.4% were obtained for the old-type TFE-coated glass fiber filter type (Pallflex "Fiberfilm" T60A20).

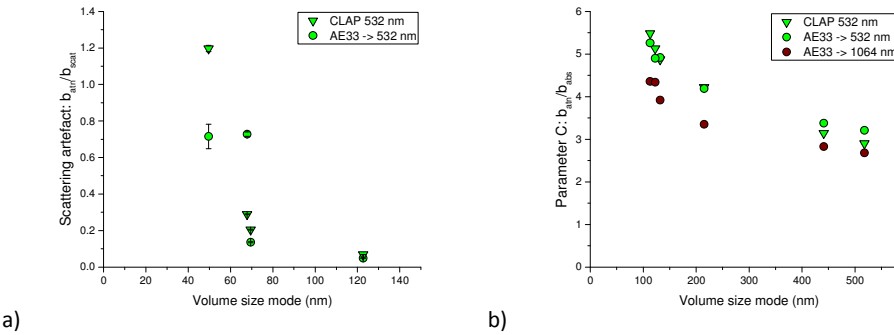

a)                                                  b)

**Figure 14. Scattering artefact of filter photometers the CLAP and the AE33 for different sizes of**

**ammonium sulfate particles (a). Multiple scattering parameter C for different sizes of soot (b). For most measurements, the standard error was smaller than the symbol.**

Similarly to scattering, there is also a strong dependence of the filter photometer response on the size of absorbing particles. The enhancement of light absorption in the filter matrix relative to particles suspended in air is described by the multiple-scattering parameter C (Weingartner et al.,

2003). For attenuation coefficient compensated for filter loading effect it holds that $C=b_{atn}/b_{abs.}$ The AE33 data was extrapolated to 532 and 1064 nm to match the PTAAM-2λ wavelengths, whereas for the CLAP only the 532 nm signal was analysed. For both filter photometers the parameter C for soot samples (Section 2.3) of different sizes shows higher absorption enhancement for smaller particles compared to larger ones (Figure 14.b). The dependence of the parameter C on the size is quite steep

at lower and atmospherically relevant sizes, then it levels off. For the AE33, the parameter C decreases at 532 nm from 5.3 to 3.2, and at 1064 nm from 4.4 to 2.7 when the volume size mode increases from 100 nm to 500 nm. These values are higher compared to the values between 2.8 and 3.6 obtained by Bernardoni et al. (2020), Yus-Díez et al. (2021) and Ferrero et al. (2021) at 520 and 660 nm. Similarly to our results, Bernardoni and Ferrero observed higher values of the parameter C

at shorter wavelengths. The dependence of the parameter C on the wavelength results in erroneous determination of the absorption Ångström exponent in filter photometers – reporting about 0.17 – 0.3 higher values compared to the PTAAM-2λ (Table 3).

**Table 3: the AE33 systematically overestimates the absorption Ångström exponents (AAE) compared to the PTAAM-2λ.**

| Campaign | Sample | Volume mode (nm) | AAE PTAAM-2λ 532-1064 nm | AAE AE33 370–950 nm |
|---|---|---|---|---|
| Ljubljana 2021 | Diesel soot | 123 | 1.07 | 1.26 |
| AeroTox 2020 | miniCAST soot | 160 | 0.86 | 1.16 |
| Ljubljana 2021 | Propane soot | 441 | 0.77 | 0.94 |


The dependence of the AE33 measured attenuation coefficient on the sample single scattering albedo (SSA) was tested with an external mixture of propane soot (441 nm volume mode) and ammonium sulphate (123 nm volume mode). The experiment was performed by filling a barrel with soot at the start, followed by continuous injection of ammonium sulphate particles (Figure 15). SSA
increased from 0.25 to 0.997. There is an increase in attenuation ratio $batn_{532nm}/babs_{532nm}$ shortly after the start of the experiment, but after the filter change (at 14:28) the ratio returns to the initial value, which indicates that the filter loading compensation was not working optimally for the selected sample. The second filter spot loading was lower and did not affect the measurement – the attenuation ratio shows a big increase for SSA>0.995 where the AE33 signal is more than doubled.
This effect has been observed in ambient measurements when the AE33 was compared to a MAAP in three different sites (Yus-Díez et al., 2021). The increase of the ratio was more than three-fold at regional background sites during the periods of high SSA.

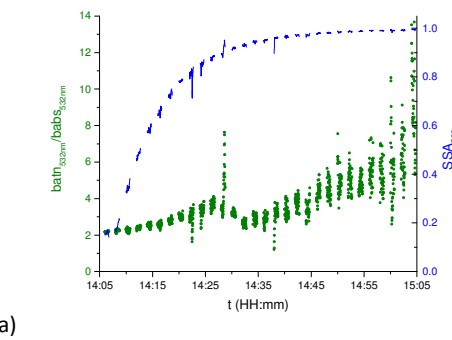
a)

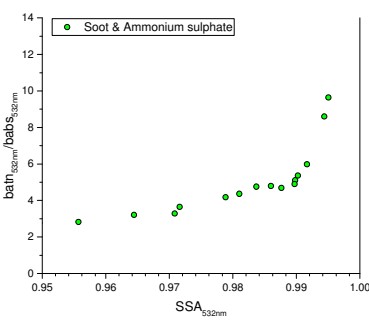
b)

**Figure 15. Measurement of externally mixed diesel soot and ammonium sulphate. The ratio**
**between the AE33 attenuation coefficient and the PTAAM-2λ absorption coefficient and single-scattering albedo as a function of time (a). There was a filter change for the AE33 at 14:28. Attenuation to absorption ratio as a function of the single-scattering albedo (b).**


**5.5 Winter ambient campaign**

The winter campaign was carried out in Ljubljana (Slovenia) in February and March 2020 to evaluate the PTAAM-2λ performance for ambient measurements. Absorption above 30 $Mm^{-1}$ at 532 nm was measured regularly during the night (Figure 16). During the day the absorption was lower, making it difficult to determine the Ångström exponent with a high time resolution – as the instrument was operating with a higher than normal noise level. Comparisons with the AE33 show similar parameter

C values (Figure 16c-d) as obtained for larger particles during the laboratory campaign. Ångström exponents in the range between 0.8 and 1.4 have been obtained, with the AE33 providing 0.3 higher AAE values. For more precise ambient measurements the instrument noise at 1 s time resolution should be reduced to 1 $Mm^{-1}$ or lower.

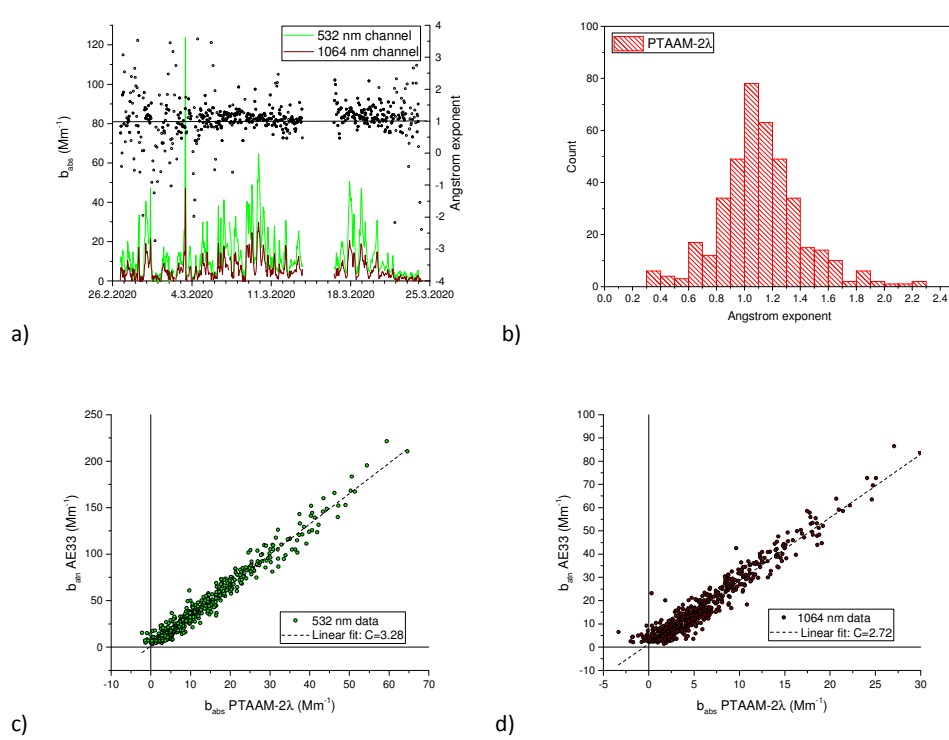

a)  b)  665  c)  d)

**Figure 16.** Absorption coefficient and Ångström exponent measured during the Ljubljana 2020 winter campaign (a) and a frequency distribution of the Ångström exponent (b). Correlations between attenuation coefficient and absorption coefficient for the 532 nm channel (c) and the 1064 nm channel (d).






**5.6 Uncertainty analysis**

The uncertainty of the measured absorption coefficient results from the calibration, method and instrumental uncertainties. The calibration of the 532 nm channel depends strongly on the uncertainty of $NO_2$ amount fraction in the calibrating gas mixture. The standard uncertainty of $NO_2$ amount fraction produced by TMPG was below 2%. The calibration of the 1064 nm channel depends both on the uncertainty of the 532 nm channel and the uncertainty of the nigrosin absorption ratio

$b_{abs,1064\,nm}/b_{abs,532\,nm}$; this parameter depends on the correct determination of the nigrosin refractive index and the measured size distribution. Determination of the refractive index is sensitive to the variation of nigrosin sample thickness which influence the reflectivity of thin film; resulting uncertainy of 3% is assessed. For well serviced SMPS instruments, the uncertainty is below 10% (Wiedensohler et al, 2017), the uncertainty of the ratio of the absorption coefficients is lower,

resulting in 4% uncertainty. Instrument operation can be influenced by the scattering artefact and the presence of absorbing gases. The absorption of gases is subtracted by measurement of filtered air, but small amouth of the gas can be adsorbed to the filter material. The combined uncertainty of scattering and gases of 1% is assessed. The final uncertainty contribution comes from the stability of the instrument response (3%).

Combined standard uncertainties for the determination of absorption coefficients and absorption Ångström exponent are presented in Table 4. The 1064 nm channel uncertainy (6%) is higher compared to the 532 nm channel (4%) due to the additional calibrations step with nigrosin particles. The uncertainty of the absorption Ångström exponent (9%) is higher compared to the absorption coefficients because of the properties of the logarithmic function.

**Table 4. The sources of uncertainty for PTAAM-2λ measurements and combined standard uncertainties (k=1) for obtaining absorption coefficients and absorption Ångström exponent (AAE). Combined uncertainties were calculated using independent uncertainty components.**

| | Sources of uncertainty | Uncertainty | Components |
|---|---|---|---|
| A | $NO_2$ amount fraction | 2% | |
| B | Nigrosin refractive index | 3% | |
| C | Mie calculation of $b_{abs,1064\,nm}/b_{abs,532\,nm}$ | 4% | |
| D | Scattering & absorbing gases | 1% | |
| E | Stability of instrument | 3% | |
| | **Combined uncertainties** | | |
| | $b_{abs,532nm}$ | 4% | A, D, E |
| | $b_{abs,1064nm}$ | 6% | A, B, C, D, E |
| | AAE | 9% | B, C, D, E, ln |




**6 Conclusions**

We report on the design and operation of the first dual-wavelength photothermal interferometer (PTAAM-2λ) for measurements of aerosol absorption. The instrumental design allows for simultaneous measurement of the same sample at both wavelengths.

The instrument was calibrated with $NO_2$ at 532 nm and the calibration was transferred to 1064 nm using aerosolized nigrosin. We were able to use Mie theory to calculate the ratio of the absorption coefficients at the two wavelengths due to the spherical shape of these particles. The comparison of this modelled absorption coefficient with the measured one for polydisperse nigrosin at 532 nm shows a 6% and 24% difference for the AeroTox 2020 and Ljubljana 2021 campaigns,
correspondingly. This validates our approach.

The PTAAM-2λ is one of the few instruments able to measure aerosol absorption coefficients at high SSA. We have demonstrated the long-term stability of the instrument and shown that the determination of the absorption coefficient is artefact-free. Uncertainties of 4%, 6% and 9% were determined for the absorption coefficients at 532 nm, 1064 nm and absorption Ångström exponent,
respectively. We have used the dual-wavelength photothermal interferometer to determine the artefacts of filter photometers and quantify their dependence on the wavelength and the aerosol size. The PTAAM-2λ shows lower AAE compared to filter photometers, demonstrating a wavelength dependence of the filter photometer multiple-scattering parameter C. We have also shown how the quantification of the filter photometer sensitivity on the aerosol size is determined and that for low
SSA, filter photometers perform satisfactorily if correct values of parameter C are used.

We believe that the demonstrated operation and performance makes the PTAAM-2λ a strong candidate for reference measurements of the aerosol absorption coefficient.



*Data availability.* The raw data and measurement logs are available at http://repozitorij.ung.si/Dokument.php?id=23720. Additional related data can be made available upon request.

*Competing interests.* LD, GM and UJ are or were employed by Haze Instruments d.o.o., the manufacturer of the described instrument. Technologies described here-in have been protected with patents.

*Author contributions.* LD and GM designed and developed the PTAAM-2λ. LD carried out the majority of experiments and analyzed the data. EW, BV and JR provided input into the experimental design of the photothermal interferometer. UJ performed measurements of nigrosin refractive index and laser spectra. LP performed scanning electron microscopy. MŠ performed measurements with atomic force microscope. MK performed integrating sphere measurements, KVi performed SMPS measurements during Ljubljana 2021 campaign. LF and UJ performed Mie calculations and quality control. DMK and KVa carried out experiments during AeroTox 2020 campaign. TB and CP prepared and tested Tracabel Mobile Permeation Generator of $NO_2$ for PTAAM-2λ calibration. TM and AW collaborated with nephelometer measurements and calibration. LD and GM wrote the paper with other authors providing valuable additions. All co-authors contributed to the paper discussion and revision.

*Acknowledgement.* We thank Martin Gysel (PSI) for the use of the AE33, Teledyne API and EAS Envimet Analytical Systems GmbH for the loan of the $NO_2$ CAPS. We thank AMES d.o.o. for the technical support.

*Financial support.* This research has been supported by the Swiss National Science Foundation (grant no. 200021_172649), EUROSTARS programme (IMALA, grant no. 11386), Slovenian Research Agency grants P1-0385, P1-0099 and I-0033, and the European Metrology Programme for Innovation and Research (EMPIR Black Carbon, EMPIR AeroTox).



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
