# Peer review of "A dual-wavelength photothermal aerosol absorption monitor: design, calibration and performance"

_Atmospheric Measurement Techniques, 2022_

## Author Response (AR1)

**amt-2022-21 - Answer to referee #2**

We thank the referee for her/his comments which have enabled us to improve the manuscript.

Atmos. Meas. Tech. Discuss., referee comment RC1
https://doi.org/10.5194/amt-2022-21-RC1, 2022
**Comment on amt-2022-21, Drinovec et al. 2022**

Anonymous Referee #2

Referee comment on "A dual-wavelength photothermal aerosol absorption monitor: design, calibration and performance" by Luka Drinovec et al., Atmos. Meas. Tech. Discuss., https://doi.org/10.5194/amt-2022-21-RC1, 2022

The manuscript "Dual-wavelength photothermal aerosol absorption monitor" by Drinovec et al. describes a standalone and commercially viable aerosol absorption monitor based on photothermal interferometry (PTI). The manuscript outlines a thorough laboratory characterization of a new instrument, the PTAAM. The laboratory characterization includes an interesting concept for calibration. The manuscript also includes a brief characterization of various studies applying the PTAAM. The various studies fall into two categories, first the calibration of filter-based absorption monitors using PTAAM as reference and the deployment of the PTAAM at a few measurement campaigns.

Overall this manuscript is of high quality and should be published in AMT. However, I have a few comments that should be addressed first. Most of these are requests for clarification or a few sentences of discussion. The exception is the filter photometer discussion. This discussion felt rushed and incomplete, and many experimental details are introduced which were not in Methods. Moreover, it does not really belong in a paper which presents a new absorption instrument with potential applications far beyond filter photometers. For these reasons, I recommend that Section 5.4 be extracted out into a brief technical note.

**Author's response:** We strongly believe that the direct in-situ measurements are the only possible reference methods for determination of aerosol absorption coefficient. Photo-acoustic and photo-thermal interferometric instrumentation has been shown in the past to be precise and, more importantly, accurate. Measuring the influence of absorbing aerosols on the (regional) climate cannot be performed satisfactorily without reference methods. The vast majority of field instrumentation for the determination of the aerosol absorption coefficient are filter photometers – Aethalometers, MAAPs and PSAPs/CLAPs dominate in the formal and ad-hoc networks. Calibrating their response to convert the measurement of the attenuation coefficient into absorption, and the determination of limitations of filter-based measurements are crucial. This is especially so at (regional) background measurement sites, where climate change is monitored, and where measurements are taken at high SSA – this is a limitation for filter-photometers (Yus-Díez et al., 2021) and also for extinction-minus-scattering (Modini et al., 2021). We expanded the Introduction to highlight these facts.

Given the aforementioned considerations, we decided to keep section 5.4 in the manuscript to highlight and show the potential of the use of PTAAM-2λ as a reference for the calibration of the Aethalometer AE33 and the CLAP. We have expanded this section and added to the Methods to describe the experimental setup in more detail. Finally, a

new work on a more comprehensive comparison (chamber experiments with primary emissions from different sources and aging of these in an aerosol chamber) of direct methods (PTAAM-2λ and PAS) and their advantages over indirect methods (EMS and different filter-photometers) is in progress.

**Changes to the manuscript:** The following paragraph was added "Measuring the effect absorbing aerosols exert on the (regional) climate cannot be performed satisfactorily without reference methods. Reference instruments can be used to calibrate simpler but ubiquitous instrumentation. Reference instrumentation needs to measure the aerosol absorption coefficient directly, accurately and precisely. The vast majority of field instrumentation for the determination of the aerosol absorption coefficient are filter photometers. Calibrating their response to convert the measurement of the attenuation coefficient into absorption, and the determination of limitations of filter-based measurements are crucial. This is especially so at (regional) background measurement sites, where climate change is monitored, and where measurements are taken at high SSA – this is a limitation for filter photometers (Yus-Díez et al., 2021) and also for extinction-minus-scattering (Modini et al., 2021)."

The aerosol sources used in section 5.4 are already described in the Materials and Methods section. We have included the experiment setups for measurement campaigns in the Supplement S7.

Comments

- The authors mention photoacoustic spectroscopy (PAS) in their introduction and mention that PAS "experiences systematic biases when the sample contains semi-volatile organic coatings or water." However, PAS and PTI are closely related (cf. Moosmuller et al., 2009). PAS uses a modulated heating laser with a microphone for detection while PTI uses a modulated heating laser with an interferometer for detection. Any physical phenomenon which occurs in PAS should occur in PTI, unless there is some specific reason why not, such as differences in laser power density or modulation frequency. If there is a substantial difference, the authors should discuss it. If there is no difference, the authors should acknowledge it. I would expect evaporation can occur in PTAAM because the 2 W and 3 W laser powers used in the PTAAM are high.

Moosmüller, H., Chakrabarty, R. K., & Arnott, W. P. (2009). Aerosol light absorption and its measurement: A review. J. Quant. Spectrosc. Radiat. Transfer, 110(11), 844–878. https://doi.org/10.1016/j.jqsrt.2009.02.035

**Author's response:** We thank the reviewer for this comment. We have only briefly discussed PAS in the introduction. To address the comment above and the ones below, we have expanded the introduction and other parts of the manuscript to include more information on PAS. Both PAS and PTI are direct methods and show the differences between PTI and PAS. PAS is conceptually similar to PTI, both use optical excitation to heat the sample. The difference is that PAS employs an acoustic resonator, while PTI uses an optical resonator – the interferometer. PAS amplifies the signal in the acoustic resonator, significantly boosting the sensitivity. PAS, therefore, needs to track the acoustic resonant frequency, otherwise it can experience drastic changes in sensitivity. PTI maintains the maximal sensitivity by keeping the interferometer in the quadrature point, where the sensitivity is linear with phase and the change of phase is proportional

to the normalized difference of the two intensities measured by the two detectors in the Mach-Zehnder (Visser et al., 2020; and our study) or Jamin (Sedlacek, 2007) interferometers.

We have shown in carefully executed experiments with (coated) soot, that PTI and PAS instrumentation agree (Kalbermatter et al., 2022). This indicates that evaporation does not play a role in investigation of soot particles coated with a-pinene-derived secondary organic matter.

Potential differences between PTI and PAS could result from geometrical factors related to the interaction of the pump and probe beams in the case of PTI, and pump beam and the acoustic wave in PAS. As shown in Moosmüller et al. (2009), the evaporation of semi-volatile species from the particle phase reduces the sensitivity in PAS for particles larger than a few micrometers using typical photoacoustic design/operating parameters (we have not found the details in other PAS publications for a similar evaluation). The potential underestimation of the PAS relative to PTI could be due to the fact that the detection is performed in the flowing medium and that the sensitivity of the acoustic detection to the latent heat used/produced in volatilization/condensation varies spatially through the acoustic resonator. The reduction of sensitivity due to the volatilization and the increase of the sensitivity due to the condensation might be different in the acoustic resonator, while the PTI probes the same air parcel undergoing volatilization/condensation because it remains in the probe beam. These effects are dependent on the geometry of the pumping scheme in PAS and PTI, the PAS acoustic resonator geometry, the pump modulation frequency, flow, particle size distribution, optical and thermal properties of the core and coating... We are investigating these potential differences in otherwise similar techniques.

**Changes to the manuscript:** The following text was added to the Introduction: "Photoacoustic spectrometer is conceptually similar to the PTI, both use optical excitation to heat the sample. The difference is that a PAS employs an acoustic resonator, while PTI uses an optical resonator – the interferometer. PAS needs to actively track the acoustic resonant frequency to maintain maximum sensitivity, otherwise it can experience drastic changes in sensitivity. PTI maintains the maximal sensitivity by keeping the interferometer in the quadrature point, where the sensitivity is linear with phase and the change of phase is proportional to the normalized difference of the two intensities measured by the two detectors in the Mach-Zehnder (Visser et al., 2020; and our study) or Jamin (Sedlacek, 2007) interferometers.

The photoacoustic instruments may experience systematic biases when the sample contains semi-volatile organic coatings or water. As these substances evaporate from the aerosol, the latent heat causes a reduction of the detected acoustic signal (Arnott et al., 2003; Murphy, 2009; Langridge et al., 2013). As shown theoretically in Moosmüller et al. (2009), the evaporation of semi-volatile species from the particle phase reduces the sensitivity in PAS only for particles larger than a few micrometers using typical photoacoustic design/operating parameters. We have shown in experiments with coated soot, that PTI and PAS instrumentation agree (Kalbermatter et al., 2022) and that evaporation does not play a role in investigation of soot particles coated with a-pinene-derived secondary organic matter."

- Similar to the previous comment, the discussion (e.g. line 110, 625, 735) implies that

the PTAAM is an improvement to filter photometers (which it certainly is) but does not acknowledge or discuss the available alternatives like PAS and the extinction-minusscattering method (EMS, like in the commercial CAPS PMssa). Of course, a full review is outside of the manuscript's scope but the authors can briefly mention the alternatives and point to previous reviews. I do not think the authors meant to imply that PTAAM is better than PAS and EMS. If they did, then a complete and quantitative discussion is required.

**Author's response:** We have added to the Introduction and the Discussion the comparisons with indirect methods (EMS) and other direct methods, showing why EMS (similarly to filter photometers, but for other reasons) cannot perform at high SSA (disqualifying it as a method for measurements at regional and global background sites, or for calibration of instrumentation at such sites). We also discuss PAS, which is conceptually similar to PTI, and show the differences between PTI and PAS.

**Changes to the manuscript:** The following text was added to the Introduction: "Measurements of the effect of absorbing aerosols on the (regional) climate cannot be performed satisfactorily without reference methods. Reference instruments can be used to calibrate simpler but ubiquitous instrumentation. Reference instrumentation needs to measure the aerosol absorption coefficient directly, accurately and precisely. The vast majority of field instrumentation for the determination of the aerosol absorption coefficient are filter photometers. It is thus crucial to calibrate their response when converting the measured attenuation coefficient into absorption and to determine the limitations of filter-based measurements. This is especially so at (regional) background measurement sites, where climate change is monitored and where measurements are taken at high SSA – this is a limitation for filter-photometers (Yus-Díez et al., 2021) and also for extinction-minus-scattering (Modini et al., 2021)."

The authors proposed a clever cross-calibration method based on nigrosin. I have a few questions that the authors may address in their discussion. Both of the following questions might easily be answered by the authors demonstrating how the predicted absorption coefficient would change given uncertainty in the refractive index. In other words, please quantify how RIs are "in agreement with" one another by converting them to an absorption related quantity. After this, the following 2 detailed questions might become irrelevant:

- The authors consider that the real refractive index n = 1.81 +- 0.01 agrees with the literature value of 1.78 from Bluvshtein et al. 2017. But Foster et al. (2019, authors' citation) found n = 1.6 and concluded that "the discrepancy between the current [refractive index] and different refractive indices found in the literature at 405 nm suggest that different batches of Nigrosin have different absorptivity and that Nigrosin may not be a good calibration substance at shorter visible wavelengths." (This statement was focussed on the imaginary refractive index but seems to apply here.)

**Author's response:** We performed the sensitivity analysis of the influence of refractive index values on the final determination of the absorption coefficient and included the results in the Supplement S5.6. The results are used to translate the uncertainty of the refractive index, 1% for n and 3% for k, of how it influences $b_{abs,1064nm}$ (2%).

The reviewer's comments opened a question of consistency of refractive indexes measured by different methods and research groups. To avoid this issue and to reduce the uncertainties due to possible differences of the refractive index between different batches of nigrosin, we performed new measurements of the real part of the refractive index of nigrosin at 450, 520, 633, 808 and 1064 nm. These measurements of the real part of the refractive index are on average 0.03 lower compared to Bluvstein et al. (2017). The control experiment using fused silica glass sample showed measurement uncertainty of 0.02. New values of the refractive index are now used for the Mie calculation: n(532 nm) = 1.62 + 0.223i, n(1064 nm) = 1.73 + 0.0419i. The article data was recalculated using the updated refractive index values which resulted in the increase in $b_{abs,1064\ nm}$ and decrease of the Angstrom exponent by 0.06.

**Changes to the manuscript:** Supplement S5.1 was updated with the new measurement of the refractive index: "Thin film samples of nigrosin were prepared by drying nigrosin solution N4 on microscopy slides following the procedure described in section 2.1. The sample with smooth surface was selected for measurement. The nigrosin film was thick enough that almost no light was transmitted, so that the reflection from the glass substrate did not influence the measured reflection. Measurements were conducted by changing the angle of the sample between 45 and 65 degrees and determination of the maximum reflected beam power. The real part of refractive index was determined by fitting the Fresnel equation to the measured data. Measurements of refractive index of fused silica glass (Suprasil, UQG Optics, UK) show good agreement with the specification (Figure S6) with deviation of +/- 0.01. Measured refractive index was on average 0.03 lower compared to Bluvstein et al. (2017).

[Figure]

a)

b)

c)

Figure S6. Examples of Brewster angle measurements on Fused silica glass and nigrosin film (a,b). The data was fitted with the Fresnel equation. Real part of the refractive index for fused silica and nigrosin (c) compared with literature data from Bluvstein et al. (2017) and fused silica glass specification."

**Changes to the manuscript:** Figure 7.b and a short discussion was added to the section 4.2: »Real part of the refractive index was determined by the Brewster angle measurement on nigrosin film (Supplement S5.1). Measured values are on average 0.03 lower compared to Bluvstein et al. (2017) (Figure 7.b), which is just outside the measurement uncertainty of 0.02."

**Changes to the manuscript:** Figures 8, 10, 11, 14, 16 and Table 3 and the corresponding discussions have been updated.

- The authors showed that a nigrosin film has a different RI to a nigrosin solution. I am convinced. But does a nigrosin aerosol have the same RI as the nigrosin film? In future work the authors may consider monodisperse size-resolved aerosol measurements (e.g. Bluvshtein et al., 2012 https://doi.org/10.1080/02786826.2012.700141) to better constrain the nigrosin refractive index. In the present work the authors may consider adding a few comments based on a sensitivity analysis or Monte Carlo.

**Author's response:** We agree that the calibration using monodisperse nigrosin is more accurate compared to the polydisperse approach but it requires additional instrumentation for particle size selection. We have made plans to perform these measurements in the near future.

For the non-magnetic (magnetic permeability $\mu_1/\mu_0=1$) and isotropic material the refractive index is an intrinsic property of material. For the anisotropic material, RI can

be influenced by the ordering of molecules. The nigrosin film sample used for determination of the complex part of the refractive index was 850 nm thick. This thickness is too large for the nigrosin to maintain an orientation throughout the sample (molecule size is in the range of 1 nm). To our knowledge there is no information indicating the anisotropy of nigrosin which would result in different refractive indexes in different directions. This is why we think that refractive index measured on nigrosin film is applicable for calculation of absorption coefficient with Mie theory.

- Line 645, extrapolated or interpolated? Linearly or power-law fit?

**Changes to the manuscript:** The procedure was added to the Section 2.7: »For comparison with PTAAM, the AE33 data was linearly interpolated from 520 nm and 590 nm to 532 nm. The 950 nm data was extrapolated to 1064 nm using the Aethalometer's Ångström exponent.

- Line 646, reference to Section 2.3 is wrong.

**Author's response:** The reference is changed to Section 2.4

- Line 655 missing the word 'factor'

**Author's response:** It si not clear where the word 'factor' is missing.

- Line 662, what kind of soot? What source? A brief comment would be helpful.

**Author's response:** The soot source used in the experiment is described 2 lines above: »The dependence of the AE33 measured attenuation coefficient on the sample single scattering albedo (SSA) was tested with an external mixture of propane soot (441 nm volume mode) and ammonium sulphate (123 nm volume mode).«

- Figure 16. I appreciate the intellectual honesty of presenting noisy 1-second AAEs. But the data suggest that a calculation based on longer averaging times would be more meaningful.

**Author's response:** Angstrom exponent data has been removed from Figure 16.a because the Angstrom exponent frequency distribution is already presented in Figure 16.b.

- The uncertainty discussion in Section 5.6 would be clarified with more focus on the wavelengths. I would also add individual columns to Table 4 for the 2 wavelengths, since some of the uncertainties A to E depend on wavelength/laser.

**Author's response:** The uncertainties of absorption coefficients depend mainly on the calibration process. For example, the uncertainty of $NO_2$ amount fraction influences the calibration of 1064 nm channel even though the gas does not absorb at 1064 nm. In a similar fashion, the uncertainty of the refractive index at 532 nm does not influence the absorption coefficient at 532 nm because the green channel is directly calibrated with $NO_2$. We decided to keep the current structure of Table 4.

Laser emission spectrum of DPSS lasers is very stable compared to laser diodes. The change in spectrum resulting from the temperature change of 6.3ºC (total change during the instrument warm-up) results in 0.5% change of the absorption cross-section during the calibration with $NO_2$. Small changes in measurement wavelength are even less important for aerosol measurements.

**Changes to the manuscript:** The following text was added to Section S4: »Changes of the laser spectrum during warm-up (23ºC–29.3ºC) resulted in 0.5% change in the absorption cross-section.«

**Changes to the manuscript:** the following sentence was added to Section 5.6: »Ångström exponent depends mostly on the correct determination of nigrosin refractive index at the measurement wavelengths.«

- Line 550, "Based on our experience" sounds like the authors are relying on experience other than the discussion in Section 4.3. If so, please share it.

**Author's response:** The experience is based on Ljubljana laboratory campaign 2020, where a faulty SMPS was used. As the importance of the uncertainty of SMPS measurements has already been mentioned, we decided to remove the sentence.

- The authors report lower AAEs than I expected for soot and only compare them to the biased AAEs reported by filter photometers. These AAEs are also low compared to PAS measurements and optical models. Please comment briefly in the text.

**Author's response:** The measured Angstrom exponent of soot is lower than that measured by AE33. Comparing our PTAAM measurements with existing PAS measurements is difficult because of different soot sources used in the experiments and different calibration techniques. The general impression is that Angstrom exponent from PTAAM is lower compared to PAS. The best approach would be to conduct a direct comparison of PTAAM with PAS and EMS, which will be done in the near future.

Due to the importance of Angstrom exponent measurement, we rechecked our calibration procedures and found no errors. Measured Angstrom exponent depends on the calibration with nigrosin particles (refractive index, SMPS measurements and Mie calculation) – it depends on the results of the optical model (Mie theory).

**Changes to the manuscript:** The following text was added: »PTAAM-2λ values of the Ångström exponent are about 0.1 lower compared to some model predictions (Scarnato et al., 2013). Optical models for soot are much more complicated compared to nigrosin because of its complex structure: loose agglomerates behave similarly to single monomers with an Ångström exponent close to 1, in contrast to compact agglomerates, which behave more like big homogenous particles, for which the Ångström exponent decreases with size (Liu et al., 2018). The exact value of the Ångström exponent of soot is still difficult to predict as the modelled absorption coefficients differ from measurements for up to 30% (Bond and Bergstrom, 2006, Liu et al., 2020).«

The following references were added:

Scarnato, B. V., Vahidinia, S., Richard, D. T., and Kirchstetter, T. W.: Effects of internal mixing and aggregate morphology on optical properties of black carbon using a discrete dipole approximation model, Atmos. Chem. Phys., 13, 5089–5101, https://doi.org/10.5194/acp-13-5089-2013, 2013.

Liu, C., Chung, C. E., Yin, Y., and Schnaiter, M.: The absorption Ångström exponent of black carbon: from numerical aspects, Atmos. Chem. Phys., 18, 6259–6273, https://doi.org/10.5194/acp-18-6259-2018, 2018.

Liu, F., Yon, J., Fuentes, A., Lobo, P., Smallwood, G.J. and Corbin, J.C.: Review of recent literature on the light absorption properties of black carbon: Refractive index, mass absorption cross section, and absorption function, Aerosol Sci. Technol., https://doi.org/54:1, 33-51, 10.1080/02786826.2019.1676878, 2018

Bond, T.C. and Bergstrom, R.W.: Light Absorption by Carbonaceous Particles: An Investigative Review, Aerosol Sci. Technol, 40:1, 27-67, https://doi.org/10.1080/02786820500421521, 2006

- In the final line of Conclusions the authors mention using PTAAM for reference absorption measurements. This conclusion was made without any Introduction about the requirements or need for new reference measurements. Please fix.

**Author's response:** We have substantially extended the Introduction (see the replies at the beginning) adding the description why a reference instrument is required; why indirect methods cannot qualify as a reference instrument; and the advantages and disadvantages of PAS and PTI. We have also expanded section 5.4 where PTAAM was used as the reference instrument.

**amt-2022-21 - Answer to referee #3**

We thank the referee for her/his comments which have enabled us to improve the manuscript.

Atmos. Meas. Tech. Discuss., referee comment RC2
https://doi.org/10.5194/amt-2022-21-RC2, 2022
**Comment on amt-2022-21**

Anonymous Referee #3

Referee comment on "A dual-wavelength photothermal aerosol absorption monitor: design, calibration and performance" by Luka Drinovec et al., Atmos. Meas. Tech. Discuss., https://doi.org/10.5194/amt-2022-21-RC2, 2022

Drinovec et al. provides a detailed characterization and thorough discussion on the design, calibration and performance as well as the associated uncertainty of a dual-wavelength photothermal aerosol aborption monitor in determining aerosol light absorption. The paper is well written and enough details are provided for the readers to understand the instrument and the tests performed. I have a few comments/suggestions for authors to consider. I recommend publication after these comments are addressed.

Page 5, Line132 stated "NO2, which absorbs strongly in the blue-green part of the spectrum, is a commonly used calibration gas." And immediately in Line 134: "Calibration with NO2 is very common, especially for climate studies, because it is usually performed at green wavelengths, where the solar spectrum features a maximum." The two sentences are redundant. The authors can choose to keep one.

**Author's response:** We have decided to keep the second sentence.

Page 10, Line 294 mentioned "A new photothermal interferometer configuration...". Here it's not clear as written if the new configuration is relative to what was described above or older design documented elsewhere, and the authors should clarify it.

**Changes to the manuscript:** »Our photothermal interferometer configuration utilises an axicon to focus the pump beams into the sample chamber«

Page 12, Line 359: What is the selected optimum modulation frequency? Also can the authors comment on the larger error bars in the signal response observed for the 1064 nm channel?

**Author's response:** The criteria and selected modulation frequencies are introduced one paragraph lower in the manuscript. The sentence »The optimum modulation frequency is selected by the best signal-to-noise ratio.« is removed.

**Author's response:** Larger error bars for the 1064 nm channel are observed at long modulation intervals (low modulation frequencies: 6 and 9 Hz). This may be due to less efficient beam homogenisation for 1064 nm channel. The noise levels in 90-100 Hz frequency region are similar for both channels.

Page 16, Line 441: "...which proved very stable over a very long period…". It's difficult to assess what is considered "very stable" and "very long period". The authors should add more support to this statement.

**Author's response:** The alignment stability is already discussed in Section 3.1: »During one and a half years of instrument testing (including road shipment in excess of 3000 km to the two measurement campaigns) there was no need to realign the optics.«.

**Changes to the manuscript:** We change the text at Line 441 to: »The instrumental response depends on the overlap between probe and pump beams. After beam alignment both channels need to be calibrated.«